# Thiocysteine lyases as polyketide synthase domains installing hydropersulfide into natural products and a hydropersulfide methyltransferase

Song Meng[1,4], Andrew D. Steele[1,4], Wei Yan[1,4], Guohui Pan[1], Edward Kalkreuter [1], Yu-Chen Liu[1], Zhengren Xu[1] & Ben Shen [1,2,3✉]

Nature forms S-S bonds by oxidizing two sulfhydryl groups, and no enzyme installing an intact hydropersulfide (-SSH) group into a natural product has been identified to date. The leinamycin (LNM) family of natural products features intact S-S bonds, and previously we reported an SH domain (LnmJ-SH) within the LNM hybrid nonribosomal peptide synthetase (NRPS)-polyketide synthase (PKS) assembly line as a cysteine lyase that plays a role in sulfur incorporation. Here we report the characterization of an *S*-adenosyl methionine (SAM)-dependent hydropersulfide methyltransferase (GnmP) for guangnanmycin (GNM) bio-synthesis, discovery of hydropersulfides as the nascent products of the GNM and LNM hybrid NRPS-PKS assembly lines, and revelation of three SH domains (GnmT-SH, LnmJ-SH, and WsmR-SH) within the GNM, LNM, and weishanmycin (WSM) hybrid NRPS-PKS assembly lines as thiocysteine lyases. Based on these findings, we propose a biosynthetic model for the LNM family of natural products, featuring thiocysteine lyases as PKS domains that directly install a -SSH group into the GNM, LNM, or WSM polyketide scaffold. Genome mining reveals that SH domains are widespread in Nature, extending beyond the LNM family of natural products. The SH domains could also be leveraged as biocatalysts to install an -SSH group into other biologically relevant scaffolds.

[1] Department of Chemistry, The Scripps Research Institute, 130 Scripps Way, Jupiter, FL 33458, USA. [2] Department of Molecular Medicine, The Scripps Research Institute, 130 Scripps Way, Jupiter, FL 33458, USA. [3] Natural Products Discovery Center at Scripps Research, The Scripps Research Institute, 130 Scripps Way, Jupiter, FL 33458, USA. [4] These authors contributed equally: Song Meng, Andrew D. Steele, Wei Yan. ✉email: shenb@scripps.edu

Sulfur-containing natural products are structurally diverse and biologically rich in activities. Nature's ability to incorporate sulfurs into natural products provides opportunities for chemistry and enzymology discovery[1–5]. Cysteine and methionine are often incorporated directly into peptide natural products, usually with limited downstream transformations (Supplementary Fig. 1a, b)[6–9]. The sulfur sources for many other sulfur-containing natural products cannot be readily predicted, and their incorporation often involves intricate tailoring reactions of the nascent intermediates[10–13]. For natural products containing disulfide bonds, the S–S bonds generally result from intramolecular or intermolecular oxidation of two sulfhydryl (-SH) groups (Supplementary Fig. 1c)[14–19]. To date, no biosynthetic pathway or enzyme installing an intact hydropersulfide (-SSH) group into natural products has been identified.

Leinamycin (LNM, **1**) is a sulfur-containing natural product featuring a 1,3-dioxo-1,2-dithiolane moiety (i.e., containing an S–S bond), spiro-fused to an 18-membered macrolactam ring (Fig. 1a)[20]. We previously cloned the LNM biosynthetic gene cluster (BGC) from *Streptomyces atroolivaceus* S-140 and discovered a DUF-SH didomain, i.e., domain of unknown function (DUF)-cysteine lyase (SH), within the final polyketide synthase (PKS) module-8 of LnmJ of the LNM hybrid non-ribosomal peptide synthetase (NRPS)-PKS assembly line (Fig. 1b, d, and Supplementary Fig. 2a)[21,22]. Isolation of LNM E1 (**2**, Fig. 1a), a presumed biosynthetic intermediate, from both *S. atroolivaceus* S-140 wild-type and selected mutant strains[23], led to the proposal that the LNM hybrid NRPS-PKS assembly line would play a role in installing only one of the two sulfur atoms of the dioxodithiolane moiety of **1**. It was hypothesized that an L-cysteine-polyketide adduct, potentially generated by the LnmJ-DUF domain, could be cleaved at the cysteinyl C–S bond by the LnmJ-SH domain to install an -SH group at C-3 of the LNM hybrid peptide–polyketide scaffold, affording **2** as the nascent product of the LNM hybrid NRPS-PKS assembly line (Supplementary Fig. 2b)[21–23]. When assayed with varying substrate mimics, LnmJ-SH indeed exhibited a cysteine lyase activity in vitro, albeit inefficiently (Supplementary Fig. 2c)[22]. These findings suggest that the S–S bond would have been formed by a tailoring enzyme post the NRPS-PKS assembly line catalysis. However, all attempts to identify the origin of the second sulfur atom and to establish the biosynthetic pathway from **2** to **1** have failed despite exhaustive efforts (Supplementary Fig. 2d).

To explore Nature's biosynthetic reservoir for the LNM family of natural products, we recently completed a genome mining campaign, using the LnmJ-DUF-SH didomain as a molecular beacon to identify BGCs encoding other members of the LNM family of natural products, culminating in the discovery of the guangnanmycins (GNM A, **3**, and GNM B, **4**) from *Streptomyces* sp. CB01883 and the weishanmycins (WSM A, **5** and WSM A1, **6**) from *Streptomyces* sp. CB02120-2 (Fig. 1a, b)[24,25]. The structure of **3**, in particular, is biosynthetically enlightening, containing a methyldisulfide moiety, a much simpler modification than the 1,3-dioxo-1,2-dithiolane moiety in **1**, and featuring an intact disulfide bond unlike **2**, **4**, and **5** that are characterized with a "-SH" group (Fig. 1a). The presence of the methyltransferase (MT) GnmP encoded within the GNM BGC further hints at the potential involvement of a hydropersulfide biosynthetic intermediate en route to **3**. Taken together, we reasoned that **3** would serve as a better model system than **1** to study disulfide incorporation in natural product biosynthesis.

Herein, we first report GnmP as *S*-adenosyl methionine (SAM)-dependent MT that exhibits a high substrate preference for GNM P (**7**), a hydropersulfide (Fig. 2), establishing **7** as the penultimate intermediate for the biosynthesis of **3** (Fig. 1d). We next establish **4** as the disproportionation product of **7**, combined

with **2** as the disproportionation product of the corresponding hydropersulfide congener LNM E (**8**) (Fig. 1a), revealing hydropersulfides as common biosynthetic intermediates in the biosynthesis of the LNM family of natural products (Figs. 1 and 3). We then demonstrate that GnmT-SH, LnmJ-SH, and WsmR-SH are thiocysteine lyases, rather than cysteine lyases, leading to the discovery of thiocysteine lyases as PKS domains that directly install a -SSH group into the GNM, LNM, or WSM polyketide scaffold (Fig. 4) and providing a biosynthetic model for the LNM family of natural products (Fig. 1d). We finally reveal that SH domains are widespread in Nature, extending beyond the LNM family of natural products (Fig. 1c), and the SH domains could be further leveraged as biocatalysts to install an -SSH group into other biologically relevant scaffolds (Fig. 5).

## Results and discussion

### Characterization of GnmP as a SAM-dependent MT exhibiting a high substrate preference for hydropersulfides. We first characterized GnmP as a SAM-dependent hydropersulfide MT, establishing **7** as the penultimate intermediate for the biosynthesis of **3** (Figs. 1d and 2). Comparative analysis of the *lnm* and *gnm* BGCs identified *gnmP* as a candidate encoding a SAM-dependent MT; however, bioinformatics analysis fell short of predicting GnmP as a *C*-, *O*-, *N*-, or *S*-MT, as hydropersulfide MT is unknown to date (Supplementary Fig. 6). We inactivated *gnmP* in *S*. sp. CB01883, affording the Δ*gnmP* mutant strain SB21007 (Supplementary Fig. 7) that abolished production of **3**, and complementation of the Δ*gnmP* mutation by expressing a functional copy of *gnmP* in trans (i.e., SB21008) restored the production of **3** (Fig. 2b). Large-scale fermentation of SB21007 followed by isolation and structural elucidation of the accumulated metabolites indeed resulted in the detection of **7**, albeit in trace quantities (Supplementary Fig. 8a, b), with the major metabolites, including the thiol **4**, isolated and characterized as degradation products of **7** due to its intrinsic instability (Supplementary Figs. 8c, 9, and 10).

We developed a chemoenzymatic method to generate the labile hydropersulfides in situ and confirmed GnmP as a SAM-dependent hydropersulfide MT by directly assaying its activity in vitro. GnmP was readily overproduced in *Escherichia coli* and purified (Supplementary Fig. 11). However, all attempts to isolate **7** directly from SB21007 or prepare **7** semisynthetically from **4** as the substrate for GnmP failed due to its intrinsic instability. We subsequently resorted to a chemoenzymatic method to generate **7** in situ, leveraging the cystine lyase activity of LnmJ-SH reported previously (Fig. 2a)[22]. Thus, by following the same protocol established previously for LnmJ-SH, we overproduced GnmT-SH, LnmJ-SH, and WsmR-SH in *E. coli* and purified them to homogeneity (Supplementary Fig. 12)[22]. We prepared the L-cysteinyl-GNM B adduct (**9**) from **4** semisynthetically (Supplementary Fig. 13). Treatment of **9** with GnmT-SH produced **7** in situ, the rapid formation of which can be quantitively followed with the thiol-trapping reagent monobromobimane (mBB, **10**) (Fig. 2c) and confirmed by structural characterization of the resultant mBB adduct (**11**) (Supplementary Fig. 10). When **9** was treated with GnmT-SH, in the presence of GnmP and SAM, a concentration-dependent consumption of **9** was observed with concomitant production of **3**, the identity of which was established by comparison with an authentic standard (Fig. 2c).

We determined the strong substrate preference of GnmP for **7** over its thiol congener **4** by directly comparing its kinetic parameters with **7** or **4** as a substrate. We first showed that the three SH domains were all catalytically competent (Supplementary Fig. 14) and, under the optimized assay conditions, GnmT-SH, LnmJ-SH, or WsmR-SH-catalyzed production of **7** from **9**

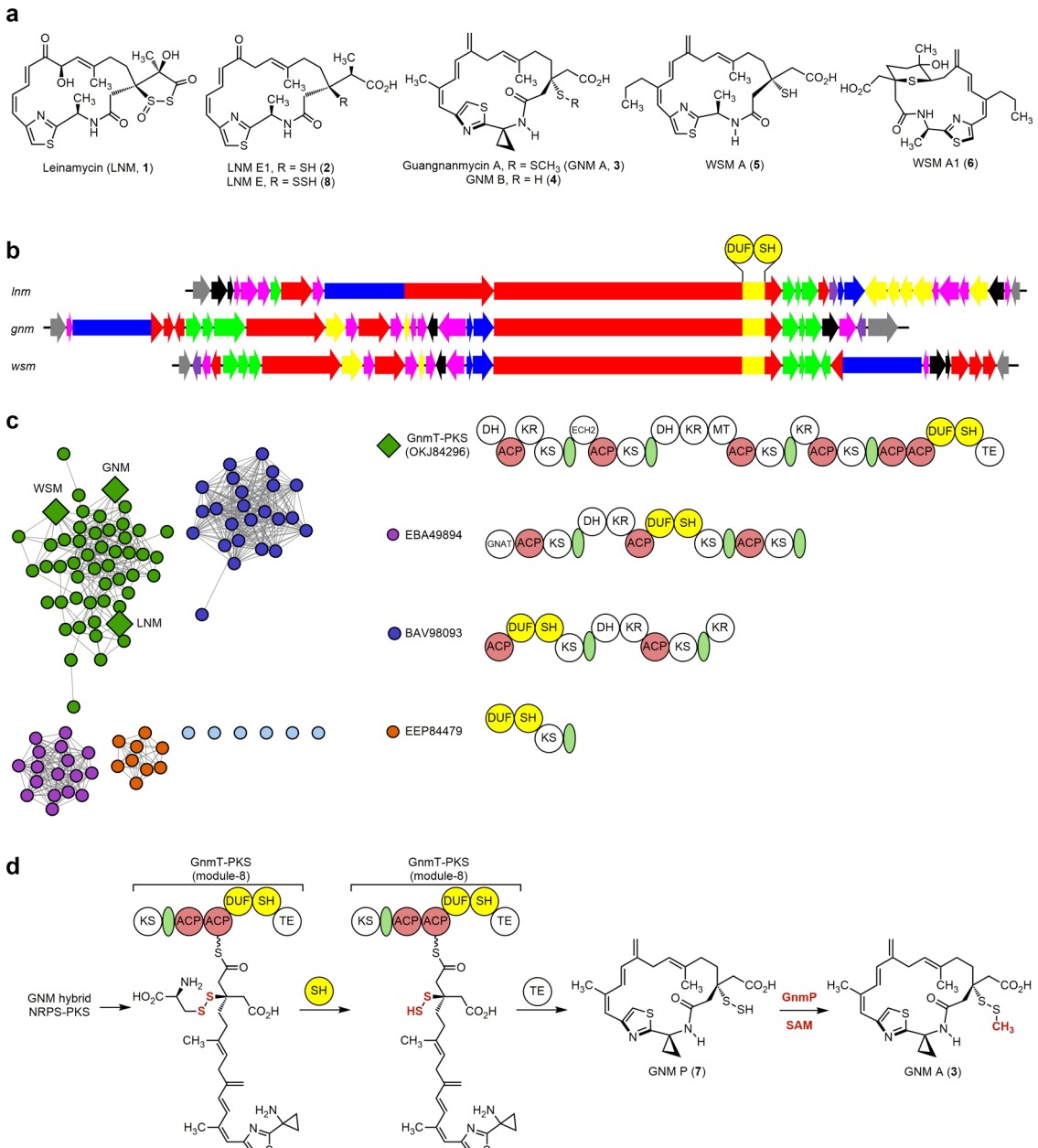

**Fig. 1 GNM biosynthesis as a model for the LNM family of natural products featuring a thiocysteine lyase as a PKS domain that directly installs an -SSH group into the GNM hybrid peptide–polyketide scaffold. a** The structures of LNM (**1**), GNM A (**3**), and congeners LNM E1 (**2**), LNM E (**8**), GNM B (**4**), WSM A (**5**), and WSM A1 (**6**) (Supplementary Fig. 3). **b** The *lnm*, *gnm*, and *wsm* BGCs featuring DUF-SH didomain-containing type I PKSs. **c** Sequence similarity network (SSN) of 109 DUF-SH domain-containing PKS proteins revealing four major clusters, with representative PKS modular architectures shown for the four major clusters (see Supplementary Figs. 4 and 5 for a complete summary). The DUF-SH didomain-containing PKS proteins from the LNM-type BGCs are color-coded in green, with the corresponding PKSs from the LNM, GNM, and WSM BGCs depicted as diamonds. **d** The biosynthetic pathway of **3** highlighting (i) GnmT-SH domain-catalyzed installation of the -SSH group via an L-thiocysteine adduct at C-3 of the growing polyketide chain, (ii) hydropersulfide **7** as the nascent product of the GNM hybrid NRPS-PKS assembly line, and (iii) GnmP-catalyzed *S*-methylation of **7** to afford **3**. ACP acyl carrier protein, DH dehydratase, DUF domain of the unknown function, ECH2 enoyl-CoA hydratase homolog 2, GNAT GCN5-related *N*-acetyltransferase, KR ketoreductase, KS ketosynthase, MT methyltransferase, SH thiocysteine lyase, TE thioesterase, and the green ovals denoting an acyltransferase docking domain.

followed Michaelis–Menten kinetics[22]. By comparing their steady-state kinetic parameters (Supplementary Fig. 15), we revealed that WsmR-SH exhibited the highest catalytic efficiency ($k_{cat}/K_m$) in generating **7** from **9**, among the three SH domains tested. This allowed us to develop a WsmR-SH and GnmP-coupled assay for kinetic analysis of GnmP using **9** as a surrogate substrate to circumvent the intrinsic instability of **7** (Supplementary Fig. 16a). By generating **7** in situ using an excess of WsmR-

SH (220 μM) relative to GnmP (50 nM), we showed that GnmP-catalyzed formation of **3** followed Michaelis–Menten kinetics and determined the steady-state kinetic parameters, with a $K_m$ value of 2.45 ± 0.41 μM and a $k_{cat}$ value of 9.53 ± 0.36 min$^{-1}$ (Supplementary Fig. 16b). GnmP could also catalyze *S*-methylation of **4**, but very inefficiently (Supplementary Figs. 16c, d, e, and 17a), and steady-state kinetic parameters of GnmP-catalyzed *S*-methylation of **4** were similarly determined, with a $K_m$ value of 12.1 ± 2.2 μM

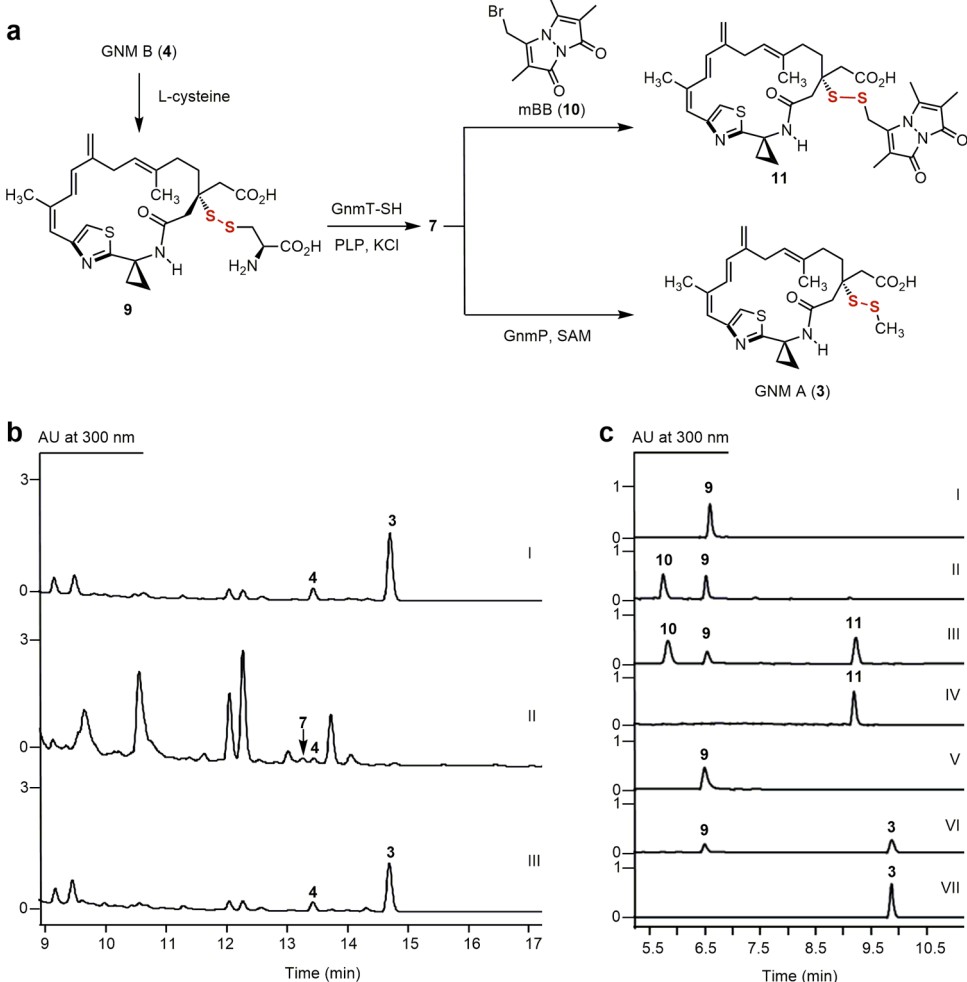

**Fig. 2 In vivo and in vitro characterization of GnmP as a SAM-dependent hydropersulfide methyltransferase, establishing 7 as the penultimate intermediate for 3 biosynthesis. a** Chemoenzymatic synthesis of **7** from **4** and a GnmT-SH and GnmP-coupled assay of GnmP-catalyzed *S*-methylation of **7** to **3** using **9** as a surrogate substrate. **b** HPLC analysis of metabolite profiles: (I) *S.* sp. CB01883 (wild-type), (II) SB21007 (Δ*gnmP*), and (III) SB21008 (Δ*gnmP/gnmP*). **c** HPLC analysis of in vitro assays: (I) substrate **9**, (II) **9** + GnmT-SH (boiled) + **10**, (III) **9** + GnmT-SH + **10**, (IV) product **11**, (V) **9** + GnmT-SH (boiled) + GnmP (boiled) + SAM, (VI) **9** + GnmT-SH + GnmP + SAM, and (VII) product **3**. Substrate and enzyme concentrations used: **9**, 0.5 mM; SAM, 2 mM; GnmT-SH, 10 μM; and GnmP, 30 μM. Incubation time: 20 min. The concentration of **10** used to trap **7** in situ as **11**: 0.5 mM. "AU" denotes absorbance units.

and a $k_{cat}$ value of $(4.36 \pm 0.29) \times 10^{-3}\,\text{min}^{-1}$ (Supplementary Fig. 16f). GnmP exhibits more than 10,000-fold higher catalytic efficiency for **7** ($k_{cat}/K_m = 3.9\,\mu\text{M}^{-1}\,\text{min}^{-1}$) over **4** ($k_{cat}/K_m = 3.6 \times 10^{-4}\,\mu\text{M}^{-1}\,\text{min}^{-1}$), in agreement with the in vivo results that support **7** as the penultimate intermediate for the biosynthesis of **3** (Fig. 2c).

**Establishment of hydropersulfide intermediates in the biosynthesis of the LNM family of natural products.** The finding of GnmP as a SAM-dependent hydropersulfide MT in the biosynthesis of **3**, combined with our inability to experimentally verify **2** as an intermediate in LNM biosynthesis (i.e., conversion of **2** to **1**, Supplementary Fig. 2d), prompted us to investigate hydropersulfides as common intermediates in the biosynthesis of the LNM family of natural products. Since hydropersulfides are known to undergo disproportionation, affording a thiol and a hydrotrisulfide that can undergo further decomposition[26], the intrinsic instability of **7** promoted us to ask if **4** is a disproportionation product of **7**, rather than a true intermediate, in the biosynthesis of **3**. In situ generation of **7** by treating **9** with an excess of WsmR-SH indeed resulted in detectable quantities of **4**,

and when this experiment was repeated with the addition of GnmP and SAM, the *S*-methylated trisulfide product (**12**) of **4** was also observed (Fig. 3a, b and Supplementary Fig. 17b). The L-cysteinyl-LNM E1 adduct (**13**) was then similarly prepared from **2** as a substrate to prepare the corresponding hydropersulfide **8** chemoenzymatically. Treatment of **13** with an excess of WsmR-SH produced **8** in situ, the identity of which was confirmed by structural characterization of its mBB adduct (Supplementary Figs. 10 and 18a, b). When the same experiment was repeated with the addition of GnmP and SAM, the *S*-methylated trisulfide product (**15**) was indeed observed, together with **2**, as well as the *S*-methylated disulfide product (**14**) of **8** (Fig. 3a, c and Supplementary Figs. 10 and 18a, c).

The establishment of **7** as the penultimate intermediate for the biosynthesis of **3**, together with the revelation of **2** and **4** as the disproportionation products of their corresponding true hydropersulfide intermediates in the biosynthesis of **1** and **3**, led to the proposal of **7**, and hydropersulfides in general, as the nascent products of the hybrid NRPS-PKS assembly lines for the LNM family of natural products. These findings account for our failed attempts to date to identify the origin of the second sulfur atom

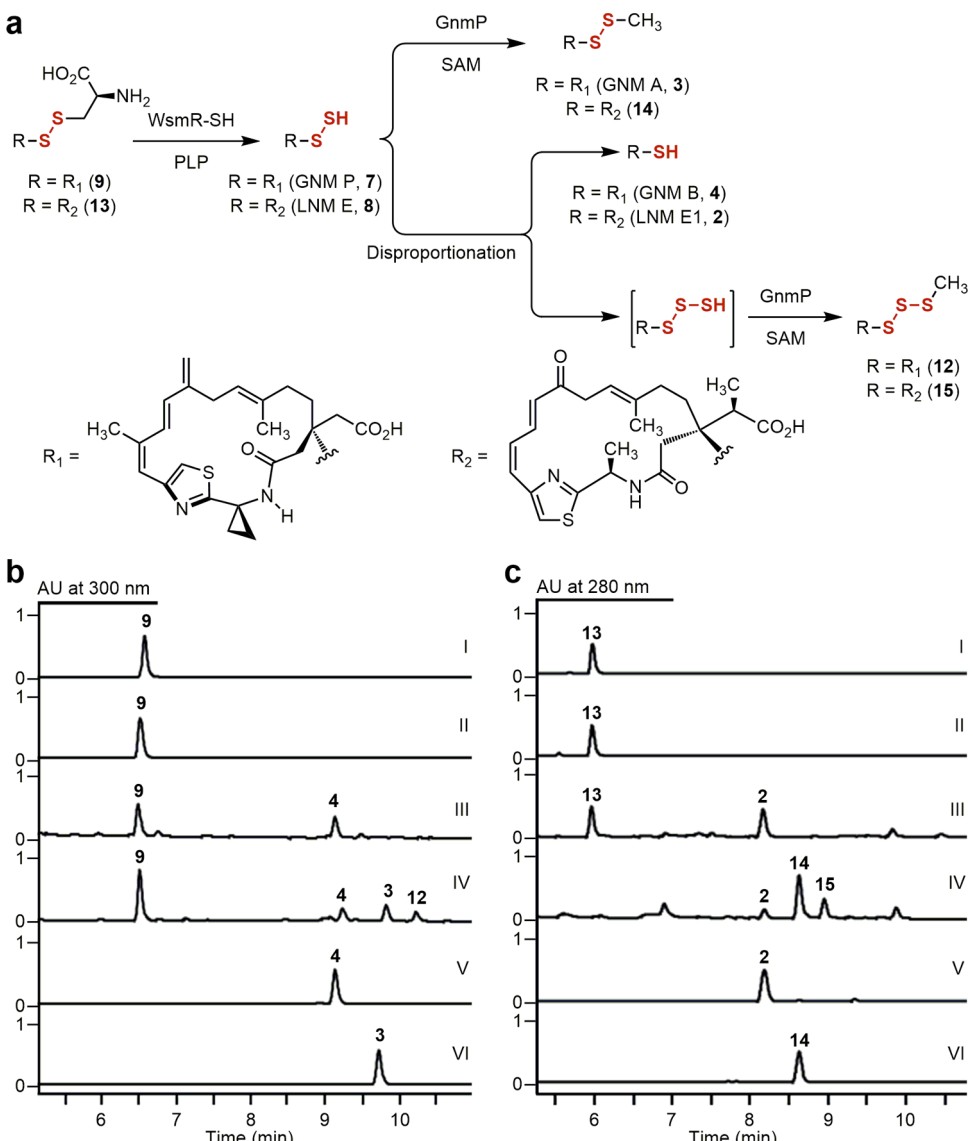

**Fig. 3 Disproportionation of GNM P (7) and LNM E (8) to GNM B (4) and LNM E1 (2), and their corresponding hydrotrisulfides, revealing 4 and 2 as shunt metabolites in GNM and LNM biosynthesis. a** Disproportionation reaction of **7** or **8**, generated in situ using **9** or **13** as a surrogate substrate, to **4** or **2** and the corresponding hydrotrisulfides, which can be trapped by GnmP in situ to afford the *S*-methylated trisulfide **12** or **15**, respectively. **b** HPLC analysis of **7** disproportionation: (I) substrate **9**, (II) **9** + WsmR-SH (boiled), (III) **9** + WsmR-SH, (IV) **9** + WsmR-SH + GnmP + SAM, (V) standard **4**, and (VI) standard **3**. Substrate and enzyme concentrations used: **9**, 1 mM; SAM, 2 mM; WsmR-SH, 220 μM; and GnmP, 10 μM. Incubation time: 3 min for all assays. **c** HPLC analysis of **8** disproportionation: (I) substrate **13**, (II) **13** + WsmR-SH (boiled), (III) **13** + WsmR-SH, (IV) **13** + WsmR-SH + GnmP + SAM, (V) standard **2**, and (VI) standard **14**. Substrate and enzyme concentrations used: **13**, 5 mM, SAM, 10 mM; WsmR-SH, 220 μM; and GnmP, 10 μM. Incubation time: 3 min for assays II and III and 2 h for assay IV, respectively.

and to establish the intermediacy of **2** in **1** biosynthesis. Isolation of **2** was the major motivation in previous studies that led to the characterization of the LnmJ-SH domain as a cysteine lyase installing the -SH group at C-3 of the LNM hybrid peptide–polyketide scaffold (Supplementary Fig. 2b–d). Demonstration of **2** and **4** as disproportionation products of **8** and **7** in the biosynthesis of **1** and **3**, respectively, therefore, inspired us to consider hydropersulfides as the nascent products of the GNM, LNM, and WSM hybrid NRPS–PKS assembly line and re-evaluate the mechanism by which the -SSH groups are installed into the varying hybrid peptide–polyketide scaffolds (Fig. 1d). These findings highlight the value of GNM as a preferred model system to study disulfide incorporation in natural product biosynthesis.

**The revelation of SH domains as thiocysteine lyases directly installing an -SSH group into varying polyketide scaffolds.** We showed that GnmT-SH, LnmJ-SH, and WsmR-SH, the three SH domains from the GNM, LNM, and WSM biosynthetic machinery of the LNM family of natural products, uniformly act as thiocysteine lyases by direct assay of varying substrate mimics in vitro. Thus, in an analogy to our previous effort to assay the LnmJ-SH domain as a cysteine lyase using an *N*-acetylcysteamine thioester (SNAC) of a truncated ʟ-cysteine-polyketide adduct (**16**) as a substrate mimic[22], we chemically synthesized the SNAC of a truncated ʟ-thiocysteine-polyketide adduct (**17**) as a substrate mimic to the ACP-tethered growing hybrid peptide–polyketide intermediate (Fig. 1d) and directly assayed the three SH domains for the thiocysteine lyase activity in vitro, with **16** as a control to

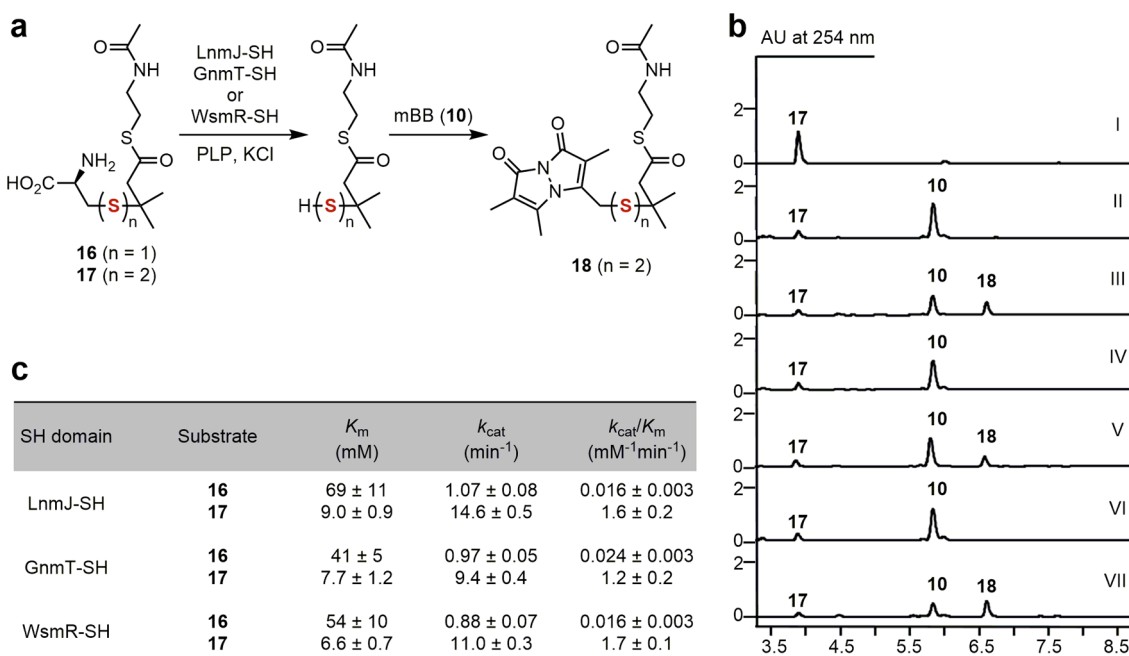

**Fig. 4 Characterization of the GnmT-SH, LnmJ-SH, and WsmR-SH domains as thiocysteine lyases installing an -SSH group into varying polyketide scaffolds. a** In vitro assay of the three SH domains as thiocysteine lyases using **17** as a substrate mimic, in comparison with **16** as a control for the cysteine lyase activity reported previously for LnmJ-SH. **b** HPLC analysis of in vitro assays: (I) substrate **17**, (II) **17** + GnmT-SH (boiled) + **10**, (III) **17** + GnmT-SH + **10**, (IV) **17** + LnmJ-SH (boiled) + **10**, (V) **17** + LnmJ-SH + **10**, (VI) **17** + WsmR-SH (boiled) + **10**, (VII) **17** + WsmR-SH (boiled) + **10**, and (VIII) **17** + WsmR-SH + **10**. Substrate and enzyme concentrations used: **17**, 5 mM; GnmT-SH, LnmJ-SH, or WsmR-SH, 30 µM. Incubation time: 30 min. The concentration of **10** used to trap the resultant hydropersulfide product in situ as **18**: 2 mM. **c** Comparison of the steady-state kinetic parameters of the three SH domains for **17** over **16**.

compare with the cysteine lyase activity reported previously for LnmJ-SH (Fig. 4a) (Supplementary Fig. 19)[22].

GnmT-SH, LnmJ-SH, and WsmR-SH indeed catalyzed cleavage of **17** to afford the corresponding hydropersulfide product, the identity of which was established through capture and characterization of the mBB-adduct **18** (Fig. 4a, b). Under the optimized conditions[22], we showed that both the cysteine lyase (towards **16**) and the thiocysteine lyase (towards **17**) activities of the three SH domains followed Michaelis-Menten kinetics and determined their steady-state kinetic parameters. GnmT-SH, LnmJ-SH, and WsmR-SH display uniformly smaller $K_m$ values and higher $k_{cat}$ values for **17** and exhibit 50- to 106-fold higher catalytic efficiencies ($k_{cat}/K_m$s) for **17** over **16** (Fig. 3c) (Supplementary Fig. 20). Taken together, these findings support the three SH domains natively functioning as thiocysteine lyases that directly install an -SSH group into the GNM, LNM, and WSM polyketide scaffolds (Fig. 1d and Supplementary Fig. 21)[24].

The thiocysteine lyase activity of the SH domains, acting on the growing ACP-tethered L-thiocysteine–polyketide adduct as a substrate, as exemplified by the proposed biosynthetic pathway for **3** (Fig. 1d) (also see Supplementary Fig. 21), requires that L-thiocysteine be available as a biosynthetic precursor for the GNM, LNM, and WSM hybrid NRPS–PKS assembly lines. One established biochemical pathway to L-thiocysteine in the primary metabolism of *Streptomyces* is via the β-elimination reaction of L-cystine catalyzed by cystathionine gamma lyase (CGL)[27]. We cloned and overproduced in *E. coli* the three homologous CGL enzymes from the LNM-, GNM-, and WSM-producing strains *S.* sp. CB01635, *S.* sp. CB01883, and *S.* sp. CB02120-2, respectively (Supplementary Fig. 22)[22]. The L-cystine lyase activity of the CGL enzymes was confirmed (Supplementary Fig. 23), demonstrating that L-thiocysteine is available in *S.* sp. CB01635, *S.* sp. CB01883, and *S.* sp. CB02120-2 to support LNM, GNM, and WSM

biosynthesis. In addition to L-thiocysteine, the deacetylated persulfide form of mycothiol, the primary thiol in *Streptomyces*[28], was also considered as a potential persulfide donor for the SH domain. While mycothiol contains an *N*-acetylated cysteine moiety that would preclude it from serving as a substrate for the SH domains by the PLP-dependent mechanism, the corresponding free amine is a known biosynthetic intermediate[29]. We, therefore, synthesized both the deacetylated mycothiol persulfide adduct and its monosaccharide analog, and assayed them as substrate mimics to undergo SH-catalyzed C–S bond cleavage in an analogous fashion to **17** in the presence of mBB (**10**) (Supplementary Fig. 24). However, no mBB-adduct **18** was observed under all assay conditions tested, ruling out mycothiol or its biosynthetic intermediates as potential persulfide donors and supporting an ACP-tethered L-thiocysteine–polyketide adduct as a substrate for the SH domains.

**DUF-SH didomain-containing BGCs widely spread in Nature that extends beyond the biosynthetic machinery for the LNM family of natural products.** Previously using the LnmJ-DUF-SH didomain as a molecular beacon to mine the genomes available from the public databases (as of March 2017), together with a genome survey of 5000 actinobacteria strains from the microbial strain collection at The Scripps Research Institute, we identified 49 BGCs predicted to encode 18 distinct members of the LNM family of natural products, from which the GNMs and WSMs were discovered[24]. Inspired by the current findings of the GnmT-SH, LnmJ-SH, and WsmR-SH domains as thiocysteine lyases, we re-examined the genomes available from the public databases. Among the total of 109 dereplicated BGCs (as of April 2021), we identified three additional families of BGCs encoding DUF-SH didomain-containing PKS proteins that are distinct from the BGCs encoding the LNM family of natural products (Fig. 1c).

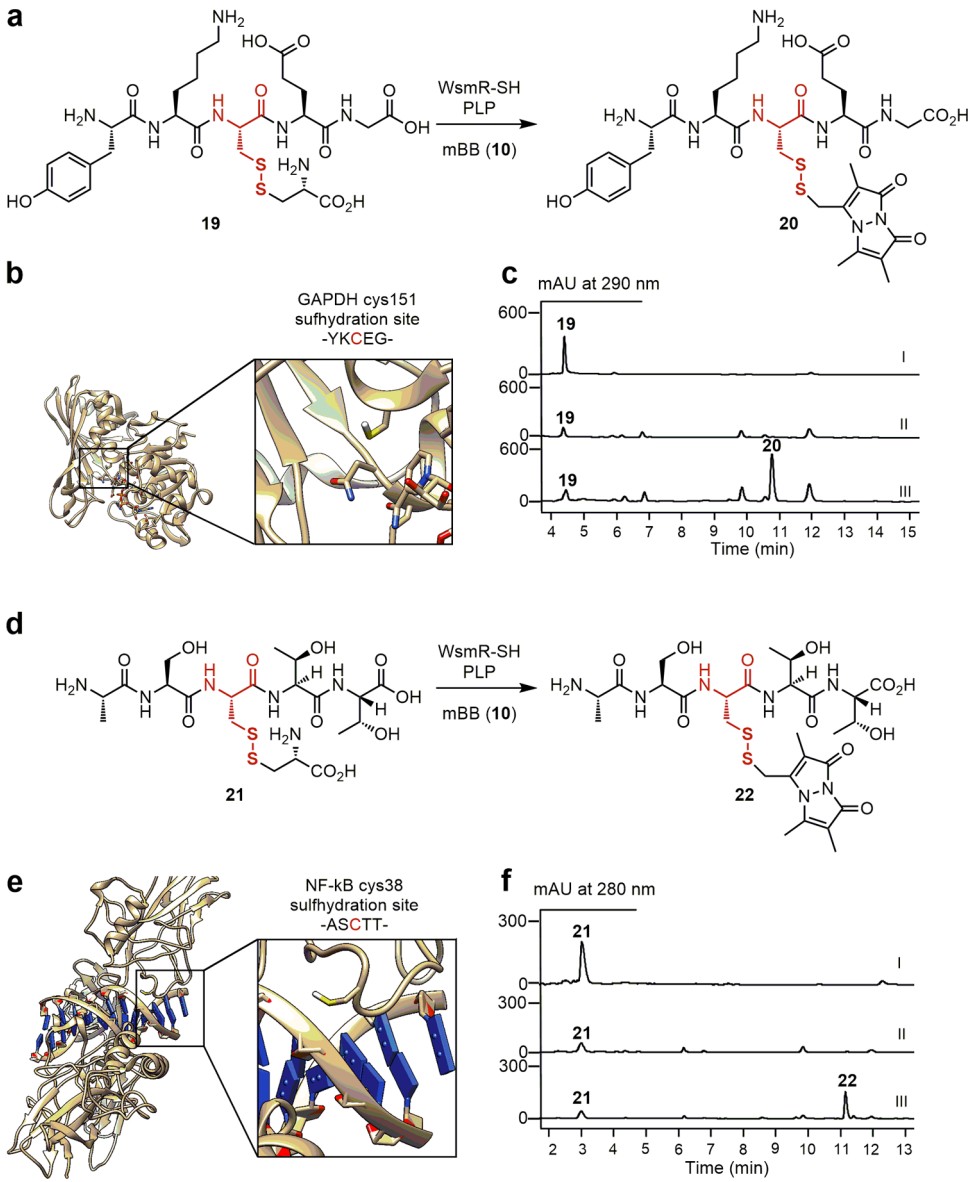

**Fig. 5 Leveraging the SH domains as biocatalysts to install an -SSH group into varying peptide scaffolds.** WsmR-SH-catalyzed synthesis of **a** 20 and **d** 22 from the L-thiocysteine-pentapeptide adducts 19 and 21 as substrate mimics for GAPDH and NF-kB, respectively. The crystal structures of **b** GAPDH (PDB: 1ZNQ) and **e** NF-kB (PDB: 1LE9) highlighting the Cys151 and Cys38 residues that undergo sulfhydration, respectively. HPLC analysis of in vitro assays of **c** (I) substrate **19**, (II) **19** + WsmR-SH (boiled) + **10**, and (III) **19** + WmR-SH + **10** and **f**, (I) substrate **21**, (II) **21** + WsmR-SH (boiled) + **10**, and (III) **21** + WmR-SH + **10**. Substrate and enzyme concentrations used: **19** or **21**, 2 mM; and WsmR-SH, 30 µM. Incubation time: 30 min. The concentration of **10** used to trap the resultant hydropersulfide product in situ as **20** or **22**, respectively: 2 mM.

While the DUF-SH didomains from both the predicted LNM-type and non-LNM type BGCs are highly homologous (Supplementary Fig. 4), the PKSs harboring the DUF-SH didomains are highly variable in their modular architecture among the varying BGCs (Fig. 1c and Supplementary Fig. 5), indicative of the biosynthesis of diverse polyketide scaffolds. These findings suggest that the SH domain chemistry discovered in the context of the biosynthesis of GNM, LNM, and WSM extends beyond the LNM family of natural products. Therefore, the DUF-SH didomains appear to be genetically programmable within type I PKS assembly lines, with the SH domain acting as a thiocysteine lyase to install an -SSH group into varying polyketide scaffolds likely through the similar chemistry as established in this study.

**Demonstration of the SH domain as a biocatalyst to install an -SSH group into peptide scaffolds.** Inspired by the substrate

promiscuity of the SH domains as thiocysteine lyases, we selected the WsmR-SH domain as a biocatalyst and demonstrated its ability to install an -SSH group into biologically relevant peptide scaffolds (Fig. 5). Hydropersulfides have emerged as an important protein post-translational modification (PTM), which are generated via the sulfhydration of cysteine residues[30]. Their roles in cellular signaling and redox biology have been increasingly appreciated[30–33], and methods to generate this reactive PTM are of interest to the scientific community, especially bioorthogonal methods that avoid the use of non-selective reagents such as $H_2S$[26]. To demonstrate the feasibility of leveraging the thiocysteine lyase activity of the SH domains to install an -SSH group into proteins, we selected regions of GAPDH and NF-kB as two model systems (Fig. 5a, d)[32,33]. We synthesized two pentapeptides, mimicking the regions surrounding the conserved Cys151 and Cys38 residues that are the sites of sulfhydration within

GAPDH and NF-kB, respectively, and conjugated them with L-cysteine to prepare the L-thiocysteine adducts **19** and **21** as substrate mimics (Fig. 5b, e). Treatment of **19** and **21** with WsmR-SH resulted in the concentration-dependent formation of new products, the identities of which as the corresponding hydropersulfides were established by analyzing their mBB adducts **20** and **22** (Supplementary Fig. 25c, d). The SH domains enrich the toolbox of excised PKS domains as biocatalysts for unparalleled chemistry with a broad substrate specificity[34–36]. The biocatalytic strategy provides a promising bioorthogonal solution to the synthesis of biologically relevant hydropersulfides with future opportunities to expand the substrate scope and improve the catalytic efficiency through enzyme evolution.

In conclusion, the LNM family of natural products has served as a great inspiration for the discovery of chemistry, enzymology, biology, and medicine[21–24]. The current study highlights how comparative analysis of the biosynthetic machinery for the LNM family of natural products has provided opportunities to study natural product biosynthesis, leading to the discoveries of a SAM-dependent MT that exhibits a high substrate preference for hydropersulfides, thiocysteine lyases as PKS domains that directly install a -SSH group into varying polyketide scaffolds, DUF-SH didomain-containing BGCs widely spread in Nature that extends beyond the biosynthetic machinery for the LNM family of natural products, and SH domains as biocatalysts that could be leveraged to install an -SSH group into varying biologically relevant scaffolds. These findings set the stage to investigate the catalytic role the DUF domains may play in the formation of the ACP-tethered L-thiocysteine–polyketide adducts and the modular architecture and programmability of the DUF-SH domain within the varying type I PKS assembly lines, as well as its portability for engineering polyketide structural diversity. Equally exciting is the continued exploration of the SH domains (as thiocysteine lyases) and GnmP (as a hydropersulfide-specific MT) as biocatalysts, alone to install an -SSH group into biologically relevant scaffolds, or in combination to cap the intrinsically unstable hydropersulfide products as methyl persulfides.

## Methods

**General experimental procedures**. All $^1H$, $^{13}C$, and 2D NMR ($^1H$-$^1H$ COSY, $^1H$-$^{13}C$ HSQC, $^1H$-$^{13}C$ HMBC, and $^1H$-$^1H$ ROESY) experiments were run on a Bruker Avance III Ultrashield 600 at 600 MHz for $^1H$ and 150 MHz for $^{13}C$ nuclei. NMR data were analyzed using MestReNova 6.0.2-5475. High-performance liquid chromatography (HPLC) was performed on an Agilent 1260 Prep Infinity LC with an MWD detector equipped with an Agilent Eclipse XDB-C18 column (250 mm × 21.2 mm, 7 µm). Analytical HPLC was performed on an Agilent 1260 Infinity LC with a DAD detector equipped with an Agilent Poroshell 120 EC-C18 column (250 mm × 4.6 mm, 2.7 µm) with a constant temperature of 35 °C.

**Liquid chromatography (LC)–high-resolution mass spectrometry (MS) analysis**. LC–MS was performed on an Agilent 1260 Infinity LC coupled to a 6230 TOF (HRESI) equipped with an Agilent Poroshell 120 EC-C18 column (50 mm × 4.6 mm, 2.7 µm) with a constant temperature of 40 °C, using mobile phase A (0.1% formic acid in $H_2O$) and mobile phase B (0.1% formic acid in $CH_3CN$). Two methods were applied for analysis. Method I was carried out using a 12 min solvent gradient from 5 to 100% B followed with 3 min of 5% B at a flow rate of 0.4 mL min$^{-1}$. Method II was carried out using a 20 min solvent gradient from 5 to 100% B followed with 5 min of 5% B at a flow rate of 0.4 mL min$^{-1}$. For the deacetylated mycothiol persulfide adduct S15 and its monosaccharide analog S16, as well as other synthetic intermediates, LC-MS was performed on a Thermo Vanquish UHPLC coupled to an Orbitrap Exploris 120 (HRESI) equipped with an Accucore C18 column (100 mm × 2.1 mm, 2.6 µm) with a constant temperature of 35 °C, using mobile phase A (0.1% formic acid in $H_2O$) and mobile phase B (0.1% formic acid in $CH_3CN$). Samples were eluted with a 5 min solvent gradient from 2 to 98% B followed with 2 min of 98% B followed with 2 min of 2% B at a flow rate of 0.4 mL min$^{-1}$.

**Bacterial strains, plasmids, and chemicals**. Strains, plasmids, and polymerase chain reaction (PCR) primers used in this study are listed in Supplementary Tables 1–3, respectively. PCR primers were purchased from Sigma-Aldrich. Q5 high-fidelity DNA polymerase, restriction endonucleases, and T4 DNA ligase were purchased from New England Biolabs (NEB) and used following the protocols

provided by the manufacturer. DNA gel extraction and plasmid preparation kits were purchased from Omega Bio-Tek. DNA sequencing was conducted by Eton Bioscience. Other chemicals, biochemical, and media components were purchased from standard commercial sources.

**Culture conditions**. E. coli strains harboring plasmids or cosmids were grown in lysogeny broth (LB) with appropriate antibiotic selection[37]. E. coli ET12567/pUZ8002 was used for intergenic conjugation with Streptomyces sp. CB01883 and the conjugations were carried out following standard procedures[38]. Streptomyces sp. CB01883 was cultivated on solid ISP4 medium for sporulation.

**Inactivation and complementation of *gnmP* in Streptomyces sp. CB01883**. To construct a plasmid for inactivation of *gnmP*, a 2063 bp DNA fragment upstream of *gnmP* was amplified with primers pOJ-1883orf131L-del-F and pOJ-1883orf131L-del-R, and a 2099 bp DNA fragment downstream of *gnmP* was amplified with primers pOJ-1883orf131R2-del-F and pOJ-1883orf131R2-del-R, using cosmid pBS21001 (1C4) of strain CB01883 as a template. The two DNA fragments were digested with the appropriate enzymes and cloned into the HindIII and EcoRI sites of pOJ260 to obtain pBS21019[39]. Then pBS21019 was transformed into E. coli ET12567 (pUZ8002) and introduced into strain CB01883 by intergeneric conjugation[38]. After several rounds of passaging the exconjugants on solid ISP4 medium, apramycin sensitive mutants were screened by PCR using primers 1883orf131ifdel-2F and 1883orf131ifdel-2R for double-crossover mutants. The genotype of the in-frame deletion mutant strain SB21007 (i.e., *ΔgnmP*) was verified by Southern analysis and PCR (Supplementary Fig. 7).

For the construction of SB21008, a 1219 bp DNA fragment containing *gnmP* was amplified with primers 1883orf131-kasO-5 and 1883orf131-kasO-3 using cosmid pBS21001 (1C4) as a template and then cloned into the SpeI and EcoRI sites of pBS21003 (pSET-KasO*: constructed by digestion of PCR products obtained with primers PSET-kasO-F and PSET-kasO-R using pSETTurdR as a template with NsiI followed by self-ligation) to obtain pBS21020, in which *gnmP* was under the control of the strong promoter *KasO*\*[24,40]. Then pBS21020 was introduced into SB21007 by intergeneric conjugation to afford the complementation strain SB21008 (i.e., SB21007/*gnmP*).

**Fermentation and identification of GNM-related metabolites**. Fresh spores of strain CB01883, its derivative mutant, and complementation strains, were individually inoculated into 250-mL baffled flasks containing 50 mL of TSB seed medium and cultured for 36 h at 28 °C and 250 rpm. For small-scale fermentations, seed culture was inoculated (10%, v/v) into 250-mL baffled flasks each containing 50 mL of production medium (soluble starch 3%, soy flour 1%, $CaCO_3$ 0.5%, $KH_2PO_4$ 0.05%, $MgSO_4$ 0.025%, $ZnSO_4·7H_2O$ 0.004%, L-methionine 0.01%, vitamin $B_{12}$ 0.0001%, pH 7.2), a medium modified from the one for the production of LNM by S. atroolivaceus S-140[20]. Amberlite XAD-16 resin (Sigma) was added to each flask (4%, w/v) at 24 h after inoculation, and the fermentation was carried out for another 96 h. The resin was harvested from the fermentation broth, washed with water, and allowed to air dry. The dry resin was extracted with MeOH (7 mL) and evaporated to dryness. The residue was suspended in $H_2O$ (4 mL) and extracted with EtOAc (7 mL × 2). The combined organic phases were evaporated to dryness, and dissolved in MeOH (7 mL), which was used directly for LC–MS analysis with Method II.

**Isolation of GNM-related metabolites S1–S5**. For isolation of GNM-related products, a large-scale fermentation (10 L) of mutant strain SB21007 was performed in a production medium. The resins were harvested and extracted with MeOH (2 L × 3). The solvent was evaporated to give the crude extract, which was fractionated with MPLC using a Biotage SNAP Ultra C18 60 g column with solvent A (0.1% formic acid in $H_2O$) and solvent B (0.1% formic acid in $CH_3CN$) as mobile phases. The crude extract was eluted using a solvent gradient from 5 to 100% B over 60 min, followed by 100% B for 5 min at a flow rate of 30 mL min$^{-1}$. The fractions containing GNM-related compounds were combined based on LC–MS analysis. Combined fractions were concentrated and subjected to a Sephadex LH-20 column eluted with methanol. The fractions containing compounds of interest were combined and finally purified by an Agilent HPLC system with a Zorbax SB-C13 column (5 µm, 9.4 × 250 mm) with solvent A (0.1% TFA in $H_2O$) and solvent B (0.1% TFA in $CH_3CN$) as mobile phases. Fraction I was eluted with a linear gradient from 20 to 65% B over 50 min at a flow rate of 2 mL min$^{-1}$ to give **S1** (4.5 mg) and **S2** (3.1 mg). Fraction II was purified in an isocratic condition of 45% B at a flow rate of 2 mL min$^{-1}$ for 40 min to afford **S3** (3.3 mg), **S4** (4.8 mg), and **S5** (3.6 mg) (Supplementary Fig. 9).

**Gene cloning**. To construct a plasmid for production of GnmP from Streptomyces sp. CB01883, the region coding *gnmP* was amplified by PCR from genomic DNA with Q5 DNA polymerase (NEB) following the protocol by the manufacturer using the primers 1883orf131-F and 1883orf131-R. The PCR product was purified, treated with T4 polymerase, and cloned into pBS3080 according to ligation-independent procedures to afford pBS21021. For the construction of plasmids pBS21022 and pBS21023, used to produce GnmT-SH and CB01883-CGL from Streptomyces sp. CB01883, the same procedure was followed with the primers

GnmT-SH-F and GnmT-SH-R, and CB01883-CGL-F and CB01883-CGL-R, respectively. For the construction of plasmid pBS22010 and pBS22011, used to produce WsmR-SH and CB02120-2-CGL from *Streptomyces* sp. CB02120-2, the same procedure was followed with the primers WsmR-SH-F and WsmR-SH-R, and CB02120-2-CGL-F and CB02120-2-CGL-R, respectively. For the construction of plasmid, pBS3168 was used to produce CB01635-CGL from *Streptomyces* sp. CB01635, the same procedure was followed with the primers CB01635-CGL-F and CB01635-CGL-R.

**Gene expression and protein production and purification.** To overproduce GnmP, pBS21021 was transformed into *E. coli* BL21(DE3) (Life Technologies), and the resultant recombinant strain was first grown in 2 L of LB, containing 50 μg/mL kanamycin, at 37 °C with shaking at 250 rpm. When the $OD_{600}$ reached 0.6, the culture was cooled to 4 °C. After the addition of 0.10 mM isopropyl β-D-1-thio-galactopyranoside (IPTG) to induce gene expression, the culture was then grown overnight at 18 °C with shaking at 250 rpm. To isolate the N-terminal His$_6$-tagged GnmP, the cells were harvested by centrifugation at 4000$g$ for 10 min at 4 °C, resuspended in lysis buffer (100 mM Tris, pH 8.0, containing 300 mM NaCl, 15 mM imidazole, and 10% glycerol), lysed by sonication, and centrifuged at 15,000$g$ for 30 min at 4 °C to pellet the cell debris. The supernatant was finally applied to a HisTrap column and purified by nickel affinity chromatography using an ÄKTA FPLC system (GE Healthcare Biosciences). The purified N-terminal His$_6$-tagged GnmP was desalted using a HiPrep desalting column (GE Healthcare Biosciences) in 50 mM Tris buffer, pH 7.8, containing 100 mM NaCl, and 5% glycerol, and concentrated using an Amicon Ultra-15 concentrator (Millipore). Protein concentrations were determined from the absorbance at 280 nm using a molar absorptivity constant ($\varepsilon_{280} = 97,860\ M^{-1}\ cm^{-1}$). GnmT-SH, CB01883-CGL, WsmR-SH, CB02120-2-CGL, and CB01635-CGL were produced and purified with the same procedure. For LnmJ-SH, expression of pBS3109 in *E. coli* and production and purification of LnmJ-SH to homogeneity followed published procedure[22].

**Analytical size-exclusion chromatography.** The molecular weights (MW) and the quaternary state of GnmP, GnmT-SH, LnmJ-SH, WsmR-SH, CB01883-CGL, CB02120-2-CGL, and CB01635-CGL in solution were determined by size-exclusion chromatography using a Superdex 200 16/600 column (GE Healthcare Life Sciences) connected to an ÄKTAxpress system. For GnmP, the buffer was 20 mM MOPS, pH 7.0, and for GnmT-SH, LnmJ-SH, WsmR-SH, CB01883-CGL, CB02120-2-CGL, and CB01635-CGL, the buffer was 50 mM MOPS, 50 mM KCl, and 100 mM NaCl, pH 7.5. The column was pre-equilibrated with two column volumes of buffer, and calibrated with vitamin B12 (1.35 kDa), myoglobin (17 kDa), ovalbumin (44 kDa), γ-globulin (158 kDa), and thyroglobin (670 kDa). The chromatography was carried out at 4 °C at a flow rate of 1 mL min$^{-1}$. The calibration curves of $K_{av}$ versus log(MW) for each buffer condition were prepared using the equation $K_{av} = (V_e - V_o)/(V_t - V_o)$, where $V_e$, $V_o$, and $V_t$ are the elution volume, column void volume, and total bed volume, respectively (Supplementary Figs. 11, 12, and 22a, b).

**Enzymatic reactions of GnmT-SH, LnmJ-SH, or WsmR-SH with 16 as a substrate.** Each incubation was performed in 50 mM sodium phosphate, pH 7.2, containing 6 mM **16**, 50 μM GnmT-SH, LnmJ-SH, or WsmR-SH, 0.2 mM PLP, and 20 mM KCl in a total volume of 50 μL, respectively. After incubation at 28 °C for 10 min, 100 μL of acetonitrile containing 0.1% v/v TFA was added to quench the reaction. The reaction mixture was then centrifuged at 12,000$g$ for 10 min. The LC–MS analysis was performed using mobile phase A (0.1% TFA in H$_2$O) and mobile phase B (0.1% TFA in CH$_3$CN) with a flow rate of 0.4 mL min$^{-1}$ and a 12 min solvent gradient from 5 to 100% B followed by 3 min of 5% B. This allowed direct quantification of substrate **16** and product **S10** in the assay solution, as well as the trace amount of **S11** resulted from a retro-Michael reaction upon quenching the assays for analysis (Supplementary Fig. 19).

**Formation of mBB-persulfide adducts of the varying hydropersulfide products generated in situ by enzymatic reactions of GnmT-SH, LnmJ-SH, or WsmR-SH.** For **11** (Fig. 2a, c) or **S9** (Supplementary Fig. 18a, b), each incubation was performed in 50 mM sodium phosphate, pH 8.0, containing 0.5 mM **9** or **13**, 10 μM GnmT-SH, LnmJ-SH, or WsmR-SH, 0.2 mM PLP, 20 mM KCl, and 0.5 mM mBB in a total volume of 50 μL, respectively. After incubation at 28 °C for 20 min, 50 μL of MeOH was added to quench the reaction. The reaction mixture was then centrifuged at 12,000$g$ for 10 min and the supernatant was injected and analyzed by LC–MS with Method I.

For **18** (Fig. 4a, b), each incubation was performed in 50 mM sodium phosphate, pH 7.5, containing 5 mM **17**, 30 μM GnmT-SH, LnmJ-SH, or WsmR-SH, 0.2 mM PLP, 20 mM KCl, and 2 mM mBB in a total volume of 50 μL, respectively. After incubation at 28 °C for 30 min, 50 μL of methanol was added to quench the reaction. The reaction mixture was then centrifuged at 12,000$g$ for 10 min and the supernatant was injected and analyzed by LC–MS with Method I.

For **20** (Fig. 5a, c) and **22** (Fig. 5d, f), each incubation was performed in 50 mM sodium phosphate, pH 7.5, containing 2 mM **19** or **21**, 30 μM GnmT-SH, LnmJ-SH, or WsmR-SH, 30 μM PLP, 20 mM KCl, 2 mM mBB, in a total volume of 50 μL, respectively. After incubation at 28 °C for 1 h, 100 μL of methanol was added to

quench the reaction. The reaction mixture was then centrifuged at 12,000$g$ for 10 min and the supernatant was injected and analyzed by LC–MS with Method II.

For the deacetylated mycothiol persulfide adduct **S15** and its monosaccharide analog **S16** as alternative substrate mimics, each incubation was performed in 50 mM Tris, pH 7.5, containing 1 mM substrate, 30 μM GnmT-SH, LnmJ-SH, or WsmR-SH, 0.2 mM PLP, 20 mM KCl, and 1 mM mBB in a total volume of 50 μL, respectively, with **17** as a positive control, affording full conversion to product **18**. After incubation at 28 °C for 20 min, 50 μL of MeOH was added to quench the reaction. The reaction mixture was then centrifuged at 12,000$g$ for 10 min and the supernatant was injected and analyzed by LC–MS with Method I.

**Enzymatic formation of *S*-methylated sulfides, disulfides, or trisulfides by GnmP.** For in situ generations of hydropersulfides **7** or **8** and its subsequent *S*-methylation to afford the *S*-methylated disulfide product **3** (Figs. 2a, c and 3a, b) or **14** (Fig. 3a, c and Supplementary Fig. 18a, c), a GnmT-SH, LnmJ-SH, or WsmR-SH and GnmP-coupled assay was employed using **9** or **13** as a surrogate substrate, respectively. Each incubation was performed in 50 mM sodium phosphate, pH 8.0, containing 0.5 mM **9** or **13**, 10 μM GnmT-SH, LnmJ-SH, or WsmR-SH, 0.2 mM PLP, 20 mM KCl, 30 μM GnmP, and 2 mM SAM in a total volume of 50 μL, respectively. After incubation at 28 °C for 20 min, 50 μL of methanol was added to quench the reaction. The reaction mixture was then centrifuged at 12,000$g$ for 10 min and the supernatant was injected and analyzed by LC–MS with Method I.

For *S*-methylation of **4** to afford the *S*-methylated sulfide product **S8** (Supplementary Fig. 16c, e), incubation was performed in 50 mM sodium phosphate, pH 8.0, 1 mM **4**, 30 μM GnmP, and 2 mM SAM in a total volume of 50 μL. After incubation at 28 °C for 3 h, 50 μL of methanol was added to quench the reaction. The reaction mixture was then centrifuged at 12,000$g$ for 10 min and the supernatant was injected and analyzed by LC–MS with Method I.

For *S*-methylation of the hydrotrisulfide, resulted from disproportionation of **7**, to afford the corresponding *S*-methylated trisulfide **12** (Fig. 3a, b), incubation was performed in 50 mM sodium phosphate, pH 8.0, containing 1 mM **9**, 220 μM WsmR-SH, 0.2 mM PLP, 20 mM KCl, 10 μM GnmP, and 2 mM SAM in a total volume of 50 μL. After incubation at 28 °C for 3 min, 250 μL of methanol was added to quench the reaction. The reaction mixture was then centrifuged at 12,000$g$ for 10 min and the supernatant was injected and analyzed by LC–MS with Method I.

For production and structural characterization of the *S*-methyl disulfide **14** and *S*-methyl trisulfide **15** (Fig. 3a, b), a large-scale enzymatic reaction was performed in 50 mM sodium phosphate, pH 8.0, containing 5 mM **13**, 220 μM WsmR-SH, 0.2 mM PLP, 20 mM KCl, 30 μM GnmP, and 10 mM SAM in a total volume of 10 ml. After incubation at 28 °C for 2 h, 30 ml of methanol was added to quench the reaction. The reaction mixture was then centrifuged at 12,000$g$ for 20 min and the supernatant was injected and analyzed by LC–MS with Method I. The reaction mixture was concentrated, dissolved in CH$_3$CN, and purified by HPLC. The isolation was conducted using a 35 min solvent gradient from 20 to 100% CH$_3$CN in H$_2$O containing 0.1% formic acid at a flow rate of 3 mL min$^{-1}$ to give **14** (3.3 mg) and **15** (0.3 mg).

**Monitoring GnmT-SH, LnmJ-SH, or WsmR-SH-catalyzed lyase reactions with the varying substrates by directly measuring the concomitantly produced pyruvate.** The direct measurement of pyruvate concomitantly produced from the GnmT-SH, LnmJ-SH, or WsmR-SH-catalyzed reactions with the varying substrates is based on the formation of methylquinoxalinol, after derivatization of the reaction mixture with *o*-phenylenediamine (OPD)[41,42] (Supplementary Figs. 15, 20, and 23). To 150 μL of 12 mM OPD in 3 N HCl was added 50 μL of reaction mixtures, and resulting mixtures vials were heated to 100 °C for 30 min. Proteins were precipitated by centrifugation and the supernatant was analyzed by analytical HPLC. HPLC analysis was performed using mobile phase A (0.1% formic acid in H$_2$O) and mobile phase B (0.1% formic acid in CH$_3$CN) with a flow rate of 1 mL min$^{-1}$ and a 20 min solvent gradient from 5 to 90% B followed by 6 min of 100% B, unless otherwise indicated. The UV absorption was measured at 340 nm. Standard curves were constructed by mixing 50 μL of the pyruvic acid solution instead of the reaction solution with 150 μL of 12 mM OPD in 3 N HCl with pyruvate concentrations ranging from 12.5 to 125 μM.

**Kinetic studies of GnmT-SH, LnmJ-SH, or WsmR-SH catalysis with 9, 16, 17, 19, or 21 as a substrate.** For kinetics studies of GnmT-SH, LnmJ-SH, or WsmR-SH catalysis with **9** as a substrate, each incubation was performed in 50 mM sodium phosphate, pH 8.0, (Tris-HCl, pH 8.5 for LnmJ-SH), 0.5 μM GnmT-SH, LnmJ-SH, or WsmR-SH, 0.2 mM PLP, and 20 mM KCl, with substrate concentrations varied from 0.76 mM to 9.12 mM, in a total volume of 50 μL, respectively. After incubation at 28 °C for 100 min, 150 μL of 12 mM OPD in 3 N HCl was added to the reaction, and the resulting mixtures vials were heated to 100 °C for 30 min. The reaction mixture was then centrifuged at 12,000$g$ for 10 min, and the supernatant was injected and analyzed by HPLC with the above method (Supplementary Fig. 15).

For kinetics studies of GnmT-SH, LnmJ-SH, or WsmR-SH with **16** as a substrate, each incubation was performed in 50 mM sodium phosphate, pH 7.2, containing 5 μM GnmT-SH, LnmJ-SH, or WsmR-SH, 0.2 mM PLP, and 20 mM

KCl, with substrate concentrations varied from 1.25 mM to 150 mM, in a total volume of 50 µL, respectively. After incubation at 28 °C for 10 min, 150 µL of 12 mM OPD in 3 N HCl was added to the reaction, and the resulting mixtures vials were heated to 100 °C for 30 min. The reaction mixture was then centrifuged at 12,000g for 10 min, and the supernatant was injected and analyzed by HPLC with the above method (Supplementary Fig. 20a–e).

For kinetics studies of GnmT-SH, LnmJ-SH, or WsmR-SH with 17 as a substrate, each incubation was performed in 50 mM sodium phosphate, pH 7.2, containing 5 µM GnmT-SH, LnmJ-SH, or WsmR-SH, 0.2 mM PLP, and 20 mM KCl, with substrate concentrations varied from 1.25 to 80 mM, in a total volume of 50 µL. After incubation at 28 °C for 10 min, 150 µL of 12 mM OPD in 3 N HCl was added to the reaction, and the resulting mixtures vials were heated to 100 °C for 30 min. The reaction mixture was then centrifuged at 12,000g for 10 min, and the supernatant was injected and analyzed by HPLC with the above method (Supplementary Fig. 20a, b, f–h).

For kinetics studies of WsmR-SH with 19 as a substrate, each incubation was performed in 50 mM sodium phosphate, pH 7.5, containing 30 µM WsmR-SH, 0.2 mM PLP, and 20 mM KCl, with substrate concentrations varied from 0.306 mM to 10 mM, in a total volume of 25 µL. After incubation at 28 °C for 10 min, 75 µL of 12 mM OPD in 3 N HCl was added to the reaction, and the resulting mixtures were heated to 100 °C for 30 min. The reaction mixture was then centrifuged at 12,000g for 10 min, and the supernatant was injected and analyzed by HPLC. HPLC analysis was performed using mobile phase A (0.1% formic acid in $H_2O$) and mobile phase B (0.1% formic acid in $CH_3CN$) with a flow rate of 1 mL min$^{-1}$ and a 40 min solvent gradient from 5 to 95% B followed by 6 min of 100% B. The UV absorption was measured at 340 nm (Supplementary Fig. 25a, b, e).

For kinetics studies of WsmR-SH with 21 as a substrate, each incubation was performed in 50 mM sodium phosphate, pH 7.5, containing 30 µM WsmR-SH, 0.2 mM PLP, and 20 mM KCl, with substrate concentrations varied from 0.375 mM to 20 mM, in a total volume of 25 µL. After incubation at 28 °C for 10 min, 75 µL of 12 mM OPD in 3 N HCl was added to the reaction, and the resulting mixtures were heated to 100 °C for 30 min. The reaction mixture was then centrifuged at 12,000g for 10 min, and the supernatant was injected and analyzed by HPLC with the above method given for substrate 19 (Supplementary Fig. 25a, b, f).

**Kinetic studies of GnmP with 7 or 4 as a substrate**. For kinetics studies of GnmP with 7 as a substrate, generated in situ from 9, each incubation was performed in 50 mM sodium phosphate, pH 8.0, containing 220 µM WsmR-SH, 0.2 mM PLP, 20 mM KCl, 0.05 µM GnmP, and 2 mM SAM, with substrate concentrations varied from 1.25 to 25 mM, in a total volume of 50 µL, respectively. After incubation at 28 °C for 3.5 min, 100 µL of methanol was added to quench the reaction. The reaction mixture was then centrifuged at 12,000g for 20 min, and the supernatant was injected and analyzed by HPLC. HPLC analysis was performed using mobile phase A (0.1% formic acid in $H_2O$) and mobile phase B (0.1% formic acid in $CH_3CN$) with a flow rate of 1 mL min$^{-1}$ and a 30 min solvent gradient from 5 to 90% B followed by 5 min of 100% B (Supplementary Fig. 16a, b).

For kinetics studies of GnmP with 4 as a substrate, each incubation was performed in 50 mM sodium phosphate, pH 8.0, containing 2 mM SAM, with substrate concentrations varied from 10 to 400 µM, and 2.5 µM GnmP in a total volume of 50 µL. After incubation at 28 °C for 200 min, 100 µL of methanol was added to quench the reaction. The reaction mixture was then centrifuged at 12,000 g for 20 min, and the supernatant was injected and analyzed by HPLC. HPLC analysis was performed using mobile phase A (0.1% formic acid in $H_2O$) and mobile phase B (0.1% formic acid in $CH_3CN$) with a flow rate of 1 mL min$^{-1}$ and a 40 min solvent gradient from 5 to 90% B followed by 10 min of 100% B (Supplementary Fig. 16c–f).

**Formation of the mBB-thiocysteine adduct of thiocysteine generated in a suit by enzymatic reactions of CB01883-CGL, CB02120-2-CGL, or CB01635-CGL with ʟ-cystine as a substrate**. As ʟ-cystine is notoriously insoluble, two equivalents of KOH were used with gentle heating and sonication to obtain a 0.1 M solution. Each incubation was performed in 50 mM sodium phosphate, pH 8.0, containing 10 mM cystine, 50 µM CB01883-CGL, CB02120-2-CGL, or CB01635-CGL, 0.2 mM PLP, and 20 mM KCl in a total volume of 50 µL, respectively. After incubation at 28 °C for 5 min, 50 µL of acetonitrile was added to quench the reaction. Subsequently, 1 µL of 100 mM mBB in acetonitrile was added to the quench reaction mixture, and incubation was conducted at room temperature for 5 min in the dark. The reaction mixture was then centrifuged at 12,000g for 10 min. The LC–MS analysis was performed using mobile phase A (0.1% TFA in H2O) and mobile phase B (0.1% TFA in $CH_3CN$) with a flow rate of 0.4 mL min$^{-1}$ and a 12 min solvent gradient from 5 to 100% B followed by 3 min of 5% B (Supplementary Fig. 23a–e, g).

**Kinetic studies of CB01883-CGL, CB01635-CGL, or CB02120-2-CGL with cystine as a substrate**. For kinetics studies of CB01883-CGL, CB02120-2-CGL, or CB01635-CGL with cystine as a substrate, each incubation was performed in 50 mM sodium phosphate, pH 8.0 (sodium phosphate, pH 7.5 for WSM-CGL), 10 µM CB01883-CGL, CB02120-2-CGL, or CB01635-CGL, 0.2 mM PLP, and 20 mM KCl, with substrate concentrations varied from 3 to 10 mM, in a total volume of 50 µL, respectively. After incubation at 28 °C for 15 min, 150 µL of

12 mM OPD in 3 N HCl was added to the reaction, and heated to 100 °C for 30 min. The reaction mixture was then centrifuged at 12,000 g for 10 min, and the supernatant was analyzed by HPLC. HPLC analysis was performed using mobile phase A (0.1% formic acid in $H_2O$) and mobile phase B (0.1% formic acid in $CH_3CN$) with a flow rate of 1 mL min$^{-1}$ and a 20 min solvent gradient from 5 to 90% B followed by 6 min of 100% B (Supplementary Fig. 23f).

**Statistics and reproducibility**. All enzyme assays and protein analysis experiments were verified with at least two independent enzyme preparations. Data are presented as mean values ± S.E.M. and error bars represent S.E.M. values.

**Reporting summary**. Further information on research design is available in the Nature Research Reporting Summary linked to this article.

## Data availability

All DNA sequence data used are publicly available: the LNM BGC sequence from S. atroolivaceous S-140 is accessible via GenBank under accession number AF484556, the GNM BGC sequence from S. sp. CB01883 under MF925481, and WSM BGC sequence from S. sp. CB02120-2 under MF925482. The raw data used for Supplementary Figs. 7, 11, 12, and 22 are provided in the Source Data file. All other data that support the findings of this study are available in the manuscript and the Supplementary Information. Source data are provided with this paper.

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

## Acknowledgements

We thank Kyowa Hakko Kogyo Co. Ltd., Japan for the wild-type *S. atroolivaceus* S-140 strain. This work was supported in part by NIH grant GM134954 (B.S.). A.D.S. and E.K. are supported in part by NIH Postdoctoral fellowships GM133114 and GM134688, respectively, and W.Y. is supported in part by the Nanjing Agricultural University, Nanjing, China and a scholarship from the Chinese Scholarship Council. This is paper #30035 from The Scripps Research Institute.

## Author contributions

B.S. conceived the project, S.M., A.D.S., W.Y. and B.S. designed the experiments, S.M., A.D.S., W.Y., G.P., E.K., Y.-C.L. and Z.X. executed the experiments, S.M., A.D.S., W.Y., E.K., Y.-C.L. and B.S. analyzed the results, and S.M., A.D.S., W.Y. and B.S. wrote the paper with inputs from all co-authors.

## Competing interests

The authors declare no competing interests.
