## [Peer Review File · Nature Communications]

Editorial Note

This manuscript has been previously reviewed at another journal that is not operating a transparent peer review scheme. This document only contains reviewer comments and rebuttal letters for versions considered at Nature Communications

REVIEWER COMMENTS

Reviewer #1 (Remarks to the Author):

I have been acting as the reviewer for this manuscript by Shen and co-workers and I appreciate the effort that the authors have put in into clarifying concerns raised by me and other reviewers. The work is of the highest quality, and with further improvements in assay procedures, the kinetic parameters presented in Figure 4c (should this be a table instead of a figure panel?) are without question (MM curves in SI Figure 20 are better than before, substrate is saturating). Most of all, it should be noted that the authors have moved away from trying to describe the source of the second S atom, rather restricting themselves to describing the activity of the SH domains. This, in the reviewer's opinion, faithfully captures the import of this work. Some concerns persist:

1. Authors postulate that thiocysteine is added to the growing polyketide as is demonstrated in Figure 1d and SI Figure 21. However, there is no direct evidence that thiocysteine is being added. This is a key point, and this still remains unresolved. This is not to say that the authors have not demonstrated the activity of the DUF domain which conceivably performs this function. The authors show that there is genetic potential to biosynthesize thiocysteine inside the producer bacteria and that the SH domains can process the thiocysteine adducts to generate hydropersulfides. But, as the manuscript is now directed towards the description of the activity of the SH domains, the true physiological substrate of the SH domains should be identified directly, rather than indirectly. There are other persulfides in the cell that can act in place of thiocysteine, the glutathione hydropersulfide comes to mind. Can the authors delete the L-cystine lyases from the producer bacteria and demonstrate the loss of production of the final molecules? Can the authors build a small rationalized catalog of disulfide substrates such as 17 in which the cysteine is replaced with other molecules such as glutathione and then perform competition experiments/kinetics to establish the primacy of thiocysteine adducts in this pathway?
2. Can methylation by GnmP occur while the hydropersulfide substrate is still bound to the ACP within the PKS assembly line? Or does methylation occur only when 7 has been offloaded by the TE? Note that the SH domain works on both thiotemplated substrate (such as 17) and the offloaded, more advanced substrate (such as 7). The authors have not compared which of the two is the physiological substrate and that the biosynthetic scheme as illustrated in Figure 1d is drawn the way it is purely because the SH domain is embedded within the collinear type I PKS module. But this extension cannot be made for the methyltransferase. The reviewers are encouraged to experimentally establish the timing for methyltransferase. It makes sense for the immediate conversion of the labile hydropersulfide to a more stable disulfide while it is still thiotemplated rather than offload a much more reactive hydropersulfide (which the authors have demonstrated degrades) and then convert to a stable form. Please note that in vivo deletion experiments do not establish timing of methyltransferase due to substrate promiscuity.
3. Some parts of the manuscript are presented in a rather jumbled fashion. Because the discovery of the gnm and wsm BGCs has been reported previously, Figure 1C is not needed where it is presented and should rather be moved to a later section where the bioinformatic mining of other actinogenomes is presented. As such, this bioinformatic mining seems to be a rather unnecessary add on and continues to impress upon the reviewer (and will to the readers eventually) that the DUF and SH domains travel together and that the activity of the DUF domain remains elusive. Similarly, biocatalysis on peptidic substrates remains an unnecessary add on. The reviewer would suggest that

the manuscript can be pruned and streamlined. The reviewer would suggest not presenting the final take-away of the experiment, as is done in lines 91–93 and elsewhere, before presenting the experimental design and data. Even though results and discussion are combined in a single section, discussion cannot precede the results.

4. Molecule 4 is called a hydrosulfide, lines 101 and 121. Would it not be better to call it a thiol? Hydrosulfide would be ok if 4 existed as a metal salt.

5. Line 176, polyketde should be polyketide.

Reviewer #2 (Remarks to the Author):

The issues were well addressed. No more concerns were raised by this reviewer.

Reviewers Responses

We again thank you and the reviewers during the revision process to help improve our manuscript. We have addressed each editorial requests (i.e., nr-editorial-checklist and nr-reporting-summary) and the final reviewer concern. The changes made in both the manuscript and Supporting Information, based on your suggestions and format requirements, are highlighted for your reference and are additionally outlined point-by-point below. It should be noted here that only the comment from reviewer #1 is addressed here, as reviewer #2 did not have any additional concerns.

Responses to Reviewer #1

I have been acting as the reviewer for this manuscript by Shen and co-workers and I appreciate the effort that the authors have put in into clarifying concerns raised by me and other reviewers. The work is of the highest quality, and with further improvements in assay procedures, the kinetic parameters presented in Figure 4c (should this be a table instead of a figure panel?) are without question (MM curves in SI Figure 20 are better than before, substrate is saturating). Most of all, it should be noted that the authors have moved away from trying to describe the source of the second S atom, rather restricting themselves to describing the activity of the SH domains. This, in the reviewer's opinion, faithfully captures the import of this work.

We appreciate the reviewer's feedback and believe that the suggested changes have made our manuscript much stronger.

Some concerns persist:

1. *Authors postulate that thiocysteine is added to the growing polyketide as is demonstrated in Figure 1d and SI Figure 21. However, there is no direct evidence that thiocysteine is being added. This is a key point, and this still remains unresolved. This is not to say that the authors have not demonstrated the activity of the DUF domain which conceivably performs this function. The authors show that there is genetic potential to biosynthesize thiocysteine inside the producer bacteria and that the SH domains can process the thiocysteine adducts to generate hydropersulfides. But, as the manuscript is now directed towards the description of the activity of the SH domains, the true physiological substrate of the SH domains should be identified directly, rather than indirectly. There are other persulfides in the cell that can act in place of thiocysteine, the glutathione hydropersulfide comes to mind.*

While glutathione is the major cellular thiol in eukaryotes, and many prokaryotes, such as Actinobacteria, utilize mycothiol as their major cellular thiol. Be that as it may, both glutathione and mycothiol do not contain the requisite amino group for the PLP-dependent C-S bond cleavage mechanism of the SH domains. However, a biosynthetic intermediate en route to mycothiol could be a potential substrate for SH, as it contains a free amine.

We therefore have taken the challenge to synthesize the deacetylated mycothiol persulfide adduct **S16 and its monosaccharide analogues **S15** and test them as potential substrate mimics. However, we found that neither compound was a substrate for the SH domains. No additional cellular thiols**

compatible with the SH domain's PLP mechanism could be identified. Additionally, while in vivo evidence of thiocysteine as a substrate for the SH domain would be ideal, it is not possible to eliminate thiocysteine from the cell (see below for more details).

Can the authors delete the L-cystine lyases from the producer bacteria and demonstrate the loss of production of the final molecules?

We agree that a knockout incapable of thiocysteine production would provide a key link between thiocysteine and the biosynthesis of LNM/GNM, however, there is no conceivable way to totally avoid the presence of thiocysteine in a cellular environment. Cystathionine-gamma lyase (CGL)-catalyzed cleavage of L-cystine is not the only source of thiocysteine in a cellular environment, it is simply one of the most well-characterized enzymatic routes to this metabolite. Any system containing H₂S and an oxidized form of cysteine (disulfide, sulfenic acid, etc) will contain some level of thiocysteine. For example, Motohashi and co-workers (*Nat. Commun.* **2017**, 8, 1177) demonstrated that the cysteinyl tRNA synthetase from *E. coli* produces mostly cysteine polysulfide aminoacyl tRNAs (>80%) from free cysteine. The persulfide-containing aminoacyl tRNAs or the resultant protein polysulfides can then form thiocysteine upon disproportionation with free cysteine. In addition, H₂S is a common *Streptomyces* metabolite (*Appl. Microbiol.* **1964**, 12, 46), which can form thiocysteine non-enzymatically upon reaction with various oxidized forms of cysteine. Our intention upon characterization of the CGL enzymes in our study was to simply demonstrate that the LNM-, GNM-, and WSM-producing strains have the capability to produce thiocysteine. Loss of LNM/GNM/WSM production upon knockout of *cgl* would not provide a definitive connection between thiocysteine and LNM/GNM/WSM.

Can the authors build a small rationalized catalog of disulfide substrates such as 17 in which the cysteine is replaced with other molecules such as glutathione and then perform competition experiments/kinetics to establish the primacy of thiocysteine adducts in this pathway?

There are two main issues with using glutathione adducts in place of the thiocysteine adducts used in our study: (i) The PLP-dependent mechanism of the SH domains requires a substrate amine functional group in proximity to the C-S bond to be cleaved in order to form the requisite aldimine intermediate, and (ii) glutathione is not utilized by *Streptomyces* as it is in eukaryotes and many other bacterial species, and mycothiol is used instead. There are very limited options for other biologically relevant sulfur-containing amino acid conjugates to be used for analogues of substrate **17** aside from cysteine, which was one of the main reasons that we originally proposed cysteine conjugates as the true SH substrates. We have now prepared two new substrates **S15** and **S16** and tested them substrates in the revision, ruling out mycothiol or its biosynthetic intermediates as potential persulfide donors and supporting an ACP-tethered L-thiocysteine-polyketide adduct as a substrate for the SH domains.

2. Can methylation by GnmP occur while the hydropersulfide substrate is still bound to the ACP within the PKS assembly line? Or does methylation occur only when 7 has been offloaded by the TE? Note that the SH domain works on both thiotemplated substrate (such as 17) and the offloaded, more advanced substrate (such as 7). The authors have not compared which of the two is the physiological substrate and that the biosynthetic scheme as illustrated in Figure 1d is drawn the way it is purely because the SH domain is embedded within the collinear type I PKS module. But this extension cannot be made for the methyltransferase. The reviewers are encouraged to experimentally establish the timing for methyltransfer.

The kinetic parameters of GnmP with GNM P make a very strong case for this to be the true substrate of GnmP (K_m of 2 μM), and we are not aware of any discrete methyltransferase enzymes that act upon ACP-tethered polyketide substrates. Regardless, we have tested the ability of GnmP to catalyze SAM-dependent methylation of the SNAC persulfide substrate derived from thiocysteine adduct **17** and did not observe any trace of the methyldisulfide product by LC-MS analysis. We have included this data in the rebuttal for your reference.

Figure 1. The persulfide resulting from the SH-catalyzed C-S bond cleavage of substrate **17** is not a viable substrate for GnmP. **a**, Attempted reaction of GnmP and SAM with the persulfide generated from substrate **17**. **b**, HPLC analysis of attempted methylation reactions, using a quench of CH₃I in CH₃CN in the absence of GnmP and SAM as a positive control. (I) substrate **17**, (II) **17** + GnmP (boiled) + WsmR-SH + PLP + SAM, (III) **17** + GnmP (boiled) + WsmR-SH (boiled) + PLP + SAM, (IV) WsmR-SH + SAM, (V) **17** + GnmP + WsmR-SH + PLP + SAM. (VI) **17** + WsmR-SH + PLP + CH₃I quench. **c**, HRMS of the methyl disulfide product **A** from panel (VI) of part **b**.

Reaction conditions: WsmR-SH (10 μM), PLP (0.2 mM), GnmP (30 μM), SAM (2 mM), **17** (2.5 mM), KCl (20 mM), phosphate buffer pH 7.5 (50 mM), incubated for 30 min at 28°C in a total volume of 50 μL. After incubation, 50 μL of methanol was added and the mixture was centrifuged for 10 minutes, and the supernatant was analyzed by LC-MS with method I. 5'-methylthioadenosine (MTA) was observed as an impurity present from the commercially purchased SAM used in the reactions. As a positive control, the same reaction was performed without GnmP or SAM, and the reaction was quenched after 20 minutes with 50 μL CH₃CN containing 5 mM CH₃I. The methyl disulfide product was not observed by UV or by extracting an ion chromatogram of the expected *m/z* value in the reactions containing GnmP.

It makes sense for the immediate conversion of the labile hydropersulfide to a more stable disulfide while it is still thiotemplated rather than offload a much more reactive hydropersulfide (which the authors have demonstrated degrades) and then convert to a stable form. Please note that in vivo deletion experiments do not establish timing of methyltransfer due to substrate promiscuity.

The stability of persulfides in vitro does not bear any significance to their stability in a cellular environment in vivo, it only provides a technical barrier to their study. There are many examples of unstable off-loaded intermediates in polyketide biosynthesis that are converted to stable products by tailoring enzymes, such as those produced by type II PKSs. The unstable intermediates are smoothly converted to the corresponding natural products by the requisite tailoring enzymes, and knockouts that lack the enzymes typically produce an array of shunt metabolites that are not detectable in wild-type strains that contain the tailoring enzymes.

3. *Some parts of the manuscript are presented in a rather jumbled fashion. Because the discovery of the gnm and wsm BGCs has been reported previously, Figure 1C is not needed where it is presented and should rather be moved to a later section where the bioinformatic mining of other actino- genomes is presented. As such, this bioinformatic mining seems to be a rather unnecessary add on and continues to impress upon the reviewer (and will to the readers eventually) that the DUF and SH domains travel together and that the activity of the DUF domain remains elusive.*

While our previous study did report genome mining analysis of DUF-SH didomains in the context of the family of LNM natural product biosynthesis, the analysis in this manuscript differs due to the inclusion of DUF-SH didomains in biosynthetic gene clusters beyond the LNM family. We have presented these data to show that the DUF-SH chemistry is more general than previously recognized, and while this may highlight the elusiveness of the DUF domain mechanism, the lack of a mechanism does not invalidate the apparent co-migration of the DUF and SH domains within PKSs across several families of natural product BGCs. The characterization of the DUF domain is our current focus beyond the work presented in this manuscript.

Similarly, biocatalysis on peptidic substrates remains an unnecessary add on. The reviewer would suggest that the manuscript can be pruned and streamlined.

We prefer to keep the biocatalysis section in the manuscript, however, we will remove it if the editor insists.

The reviewer would suggest not presenting the final take-away of the experiment, as is done in lines 91–93 and elsewhere, before presenting the experimental design and data. Even though results and discussion are combined in a single section, discussion cannot precede the results.

This critique is a stylistic concern, and we prefer to write our manuscripts with a topic sentence at the beginning of each paragraph, for the sake of clarity to the reader.

4. Molecule 4 is called a hydrosulfide, lines 101 and 121. Would it not be better to call it a thiol? Hydrosulfide would be ok if 4 existed as a metal salt.

We have changed the nomenclature used from hydrosulfide to thiol.

5. Line 176, polyketde should be polyketide.

We have corrected this mistake in the manuscript.

REVIEWERS' COMMENTS

Reviewer #1 (Remarks to the Author):

The authors have done a good job with the revisions and answered all queries. I have no further comments.